*Method*

# Multi-Omics Factor Analysis—a framework for unsupervised integration of multi-omics data sets

Ricard Argelaguet[1,†] ⓘD, Britta Velten[2,†] ⓘD, Damien Arnol[1] ⓘD, Sascha Dietrich[3] ⓘD, Thorsten Zenz[3,4,5] ⓘD, John C Marioni[1,6,7] ⓘD, Florian Buettner[1,8,*] ⓘD, Wolfgang Huber[2,**] ⓘD & Oliver Stegle[1,2,***] ⓘD

## Abstract

**Multi-omics studies promise the improved characterization of biological processes across molecular layers. However, methods for the unsupervised integration of the resulting heterogeneous data sets are lacking. We present Multi-Omics Factor Analysis (MOFA), a computational method for discovering the principal sources of variation in multi-omics data sets. MOFA infers a set of (hidden) factors that capture biological and technical sources of variability. It disentangles axes of heterogeneity that are shared across multiple modalities and those specific to individual data modalities. The learnt factors enable a variety of downstream analyses, including identification of sample subgroups, data imputation and the detection of outlier samples. We applied MOFA to a cohort of 200 patient samples of chronic lymphocytic leukaemia, profiled for somatic mutations, RNA expression, DNA methylation and *ex vivo* drug responses. MOFA identified major dimensions of disease heterogeneity, including immunoglobulin heavy-chain variable region status, trisomy of chromosome 12 and previously underappreciated drivers, such as response to oxidative stress. In a second application, we used MOFA to analyse single-cell multi-omics data, identifying coordinated transcriptional and epigenetic changes along cell differentiation.**

**Keywords** data integration; dimensionality reduction; multi-omics; personalized medicine; single-cell omics

**Subject Categories** Computational Biology; Genome-Scale & Integrative Biology; Methods & Resources

**Mol Syst Biol. (2018) 14: e8124**

## Introduction

Technological advances increasingly enable multiple biological layers to be probed in parallel, ranging from genome, epigenome, transcriptome, proteome and metabolome to phenome profiling (Hasin *et al*, 2017). Integrative analyses that use information across these data modalities promise to deliver more comprehensive insights into the biological systems under study. Motivated by this, multi-omics profiling is increasingly applied across biological domains, including cancer biology (Gerstung *et al*, 2015; Iorio *et al*, 2016; Mertins *et al*, 2016; Cancer Genome Atlas Research Network, 2017), regulatory genomics (Chen *et al*, 2016), microbiology (Kim *et al*, 2016) or host-pathogen biology (Soderholm *et al*, 2016). Most recent technological advances have also enabled performing multi-omics analyses at the single-cell level (Macaulay *et al*, 2015; Angermueller *et al*, 2016; Guo *et al*, 2017; Clark *et al*, 2018; Colomé-Tatché & Theis, 2018). A common aim of such applications is to characterize heterogeneity between samples, as manifested in one or several of the data modalities (Ritchie *et al*, 2015). Multi-omics profiling is particularly appealing if the relevant axes of variation are not known *a priori*, and hence may be missed by studies that consider a single data modality or targeted approaches.

A basic strategy for the integration of omics data is testing for marginal associations between different data modalities. A prominent example is molecular quantitative trait locus mapping, where large numbers of association tests are performed between individual genetic variants and gene expression levels (GTEx Consortium, 2015) or epigenetic marks (Chen *et al*, 2016). While eminently useful for variant annotation, such association studies are inherently *local* and do not provide a coherent global map of the molecular differences between samples. A second strategy is the use of kernel- or graph-based methods to combine different

1 European Molecular Biology Laboratory, European Bioinformatics Institute, Hinxton, Cambridge, UK
2 European Molecular Biology Laboratory (EMBL), Heidelberg, Germany
3 Heidelberg University Hospital, Heidelberg, Germany
4 German Cancer Research Center (dkfz) and National Center for Tumor Diseases (NCT), Heidelberg, Germany
5 Germany & Hematology, University Hospital Zurich and University of Zurich, Zurich, Switzerland
6 Cancer Research UK Cambridge Institute, University of Cambridge, Cambridge, UK
7 Wellcome Trust Sanger Institute, Hinxton, Cambridge, UK
8 Helmholtz Zentrum München–German Research Center for Environmental Health, Institute of Computational Biology, Neuherberg, Germany
  *Corresponding author. Tel: +49 89 23742560; E-mail: fbuettner.phys@gmail.com
  **Corresponding author. Tel: +49 6221 387 8823; E-mail: wolfgang.huber@embl.de
  ***Corresponding author. Tel: +49 6221 3878190; E-mail: oliver.stegle@embl.de
  †These authors contributed equally to this work

data types into a common similarity network between samples (Lanckriet *et al*, 2004; Wang *et al*, 2014); however, it is difficult to pinpoint the molecular determinants of the resulting graph structure. Related to this, there exist generalizations of other clustering methods to reconstruct discrete groups of samples based on multiple data modalities (Shen *et al*, 2009; Mo *et al*, 2013).

A key challenge that is not sufficiently addressed by these approaches is interpretability. In particular, it would be desirable to reconstruct the underlying factors that drive the observed variation across samples. These could be continuous gradients, discrete clusters or combinations thereof. Such factors would help in establishing or explaining associations with external data such as phenotypes or clinical covariates. Although factor models that aim to address this have previously been proposed (e.g. Meng *et al*, 2014, 2016; Tenenhaus *et al*, 2014; preprint: Singh *et al*, 2018), these methods either lack sparsity, which can reduce interpretability, or require a substantial number of parameters to be determined using computationally demanding cross-validation or post hoc. Further challenges faced by existing methods are computational scalability to larger data sets, handling of missing values and non-Gaussian data modalities, such as binary readouts or count-based traits.

# Results

We present Multi-Omics Factor Analysis (MOFA), a statistical method for integrating multiple modalities of omics data in an unsupervised fashion. Intuitively, MOFA can be viewed as a versatile and statistically rigorous generalization of principal component analysis (PCA) to multi-omics data. Given several data matrices with measurements of multiple omics data types on the same or on partially overlapping sets of samples, MOFA infers an interpretable low-dimensional data representation in terms of (hidden) factors (Fig 1A). These learnt factors capture major sources of variation across data modalities, thus facilitating the identification of continuous molecular gradients or discrete subgroups of samples. The inferred factor loadings can be sparse, thereby facilitating the linkage between the factors and the most relevant molecular features. Importantly, MOFA disentangles to what extent each factor is unique to a single data modality or is manifested in multiple modalities (Fig 1B), thereby revealing shared axes of variation between the different omics layers. Once trained, the model output can be used for a range of downstream analyses, including visualization, clustering and classification of samples in the low-dimensional space(s) spanned by the factors, as well as the automated annotation of factors using (gene set) enrichment analysis, the identification of outlier samples and the imputation of missing values (Fig 1B).

Technically, MOFA builds upon the statistical framework of group Factor Analysis (Virtanen *et al*, 2012; Khan *et al*, 2014; Klami *et al*, 2015; Bunte *et al*, 2016; Zhao *et al*, 2016; Leppäaho & Kaski, 2017), which we have adapted to the requirements of multi-omics studies (Materials and Methods): (i) fast inference based on a variational approximation, (ii) inference of sparse solutions facilitating interpretation, (iii) efficient handling of missing values and (iv) flexible combination of different likelihood models for each data modality, which enables integrating diverse data types such as binary-, count- and continuous-valued data. The relationship of

MOFA to previous approaches (Shen *et al*, 2009; Virtanen *et al*, 2012; Mo *et al*, 2013; Klami *et al*, 2015; Remes *et al*, 2015; Bunte *et al*, 2016; Hore *et al*, 2016; Zhao *et al*, 2016; Leppáaho & Kaski, 2017) is discussed in Materials and Methods and Appendix Table S3.

MOFA is implemented as well-documented open-source software and comes with tutorials and example workflows for different application domains (Materials and Methods). Taken together, these functionalities provide a powerful and versatile tool for disentangling sources of variation in multi-omics studies.

## Model validation and comparison on simulated data

First, to validate MOFA, we simulated data from its generative model, varying the number of views, the likelihood models, the number of latent factors and other parameters (Materials and Methods, Appendix Table S1). We found that MOFA was able to accurately reconstruct the latent dimension, except in settings with large numbers of factors or high proportions of missing values (Appendix Fig S1). We also found that models that account for non-Gaussian observations improved the fit when simulating binary or count data (Appendix Figs S2 and S3).

We also compared MOFA to two previously reported latent variable models for multi-omics integration: GFA (Leppäaho & Kaski, 2017) and iCluster (Mo *et al*, 2013). Over a range of simulations, we observed that GFA and iCluster tended to infer redundant factors (Appendix Fig S4) and were less accurate in recovering patterns of shared factor activity across views (Appendix Fig S5). MOFA is also computationally more efficient than these existing methods (Fig EV1). For example, the training on the CLL data, which we consider next, required 25 min using MOFA versus 34 h with GFA and 5–6 days with iCluster.

## Application to chronic lymphocytic leukaemia

We applied MOFA to a study of chronic lymphocytic leukaemia (CLL), which combined *ex vivo* drug response measurements with somatic mutation status, transcriptome profiling and DNA methylation assays (Dietrich *et al*, 2018; Fig 2A). Notably, nearly 40% of the 200 samples were profiled with some but not all omics types; such a missing value scenario is not uncommon in large cohort studies, and MOFA is designed to cope with it (Materials and Methods; Appendix Fig S1). MOFA was configured to combine different likelihood models in order to accommodate the combination of continuous and discrete data types in this study.

MOFA identified 10 factors (minimum explained variance 2% in at least one data type; Materials and Methods). These were robust to algorithm initialization as well as subsampling of the data (Appendix Figs S6 and S7). The factors were largely orthogonal, capturing independent sources of variation (Appendix Fig S6). Among these, Factors 1 and 2 were active in most assays, indicating broad roles in multiple molecular layers (Fig 2B). In contrast, other factors such as Factor 3 or Factor 5 were specific to two data modalities, and Factor 4 was active in a single data modality only. Cumulatively, the 10 factors explained 41% of variation in the drug response data, 38% in the mRNA data, 24% in the DNA methylation data and 24% in the mutation data (Fig 2C).

We also trained MOFA when excluding individual data modalities to probe their redundancy, finding that factors that were active

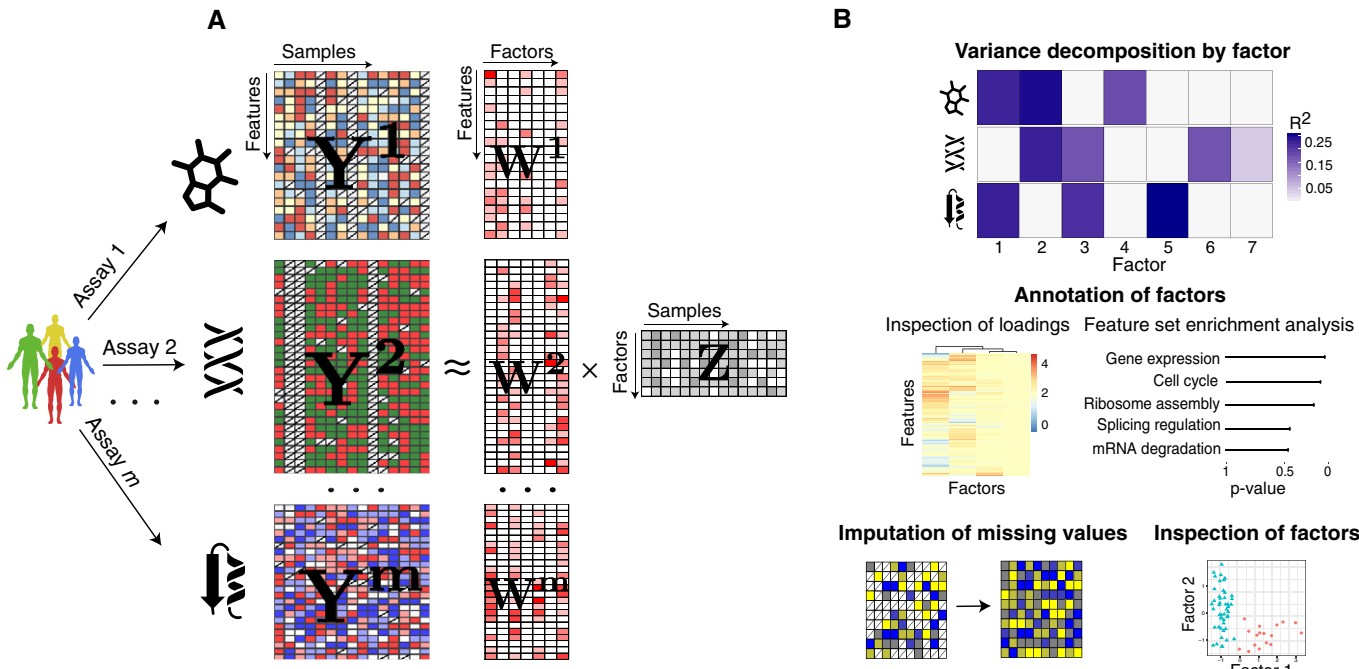

**Figure 1.   Multi-Omics Factor Analysis: model overview and downstream analyses.**

A   Model overview: MOFA takes M data matrices as input ($\mathbf{Y}^1$,..., $\mathbf{Y}^M$), one or more from each data modality, with co-occurrent samples but features that are not necessarily related and that can differ in numbers. MOFA decomposes these matrices into a matrix of factors ($\mathbf{Z}$) for each sample and $M$ weight matrices, one for each data modality ($\mathbf{W}^1$,.., $\mathbf{W}^M$). White cells in the weight matrices correspond to zeros, i.e. inactive features, whereas the cross symbol in the data matrices denotes missing values.

B   The fitted MOFA model can be queried for different downstream analyses, including (i) variance decomposition, assessing the proportion of variance explained by each factor in each data modality, (ii) semi-automated factor annotation based on the inspection of loadings and gene set enrichment analysis, (iii) visualization of the samples in the factor space and (iv) imputation of missing values, including missing assays.

in multiple data modalities could still be recovered, while the identification of others was dependent on a specific data type (Appendix Fig S8). In comparison with GFA (Leppäaho & Kaski, 2017) and iCluster (Mo *et al*, 2013), MOFA was more consistent in identifying factors across multiple model instances (Appendix Fig S9).

## MOFA identifies important clinical markers in CLL and reveals an underappreciated axis of variation attributed to oxidative stress

As part of the downstream pipeline, MOFA provides different strategies to use the loadings of the features on each factor to identify their aetiology (Fig 1B). For example, based on the top weights in the mutation data, Factor 1 was aligned with the somatic mutation status of the immunoglobulin heavy-chain variable region gene (IGHV), while Factor 2 aligned with trisomy of chromosome 12 (Fig 2D and E). Thus, MOFA correctly identified two major axes of molecular disease heterogeneity and aligned them with two of the most important clinical markers in CLL (Zenz *et al*, 2010; Fabbri & Dalla-Favera, 2016; Fig 2D and E).

IGHV status, the marker associated with Factor 1, is a surrogate of the differentiation state of the tumour's cell of origin and the level of activation of the B-cell receptor. While in clinical practice this axis of variation is generally considered binary (Fabbri & Dalla-Favera, 2016), our results indicate a more complex

substructure (Fig 3A, Appendix Fig S10). At the current resolution, this factor was consistent with three subgroup models such as proposed by Oakes *et al* (2016) and Queiros *et al* (2015) (Appendix Fig S11), although there is suggestive evidence for an underlying continuum. MOFA connected this factor to multiple molecular layers (Appendix Figs S12 and S13), including changes in the expression of genes previously linked to IGHV status (Vasconcelos *et al*, 2005; Maloum *et al*, 2009; Trojani *et al*, 2012; Morabito *et al*, 2015; Plesingerova *et al*, 2017; Fig 3B and C) and with drugs that target kinases in or downstream of the B-cell receptor pathway (Fig 3D and E).

Despite their clinical importance, the IGHV and the trisomy 12 factors accounted for < 20% of the variance explained by MOFA, suggesting the existence of other sources of heterogeneity. One example is Factor 5, which was active in the mRNA and drug response data. Analysis of the weights in the mRNA revealed that this factor tagged a set of genes enriched for oxidative stress and senescence pathways (Figs 2F and EV2A), with the top weights corresponding to heat-shock proteins (HSPs; Fig EV2B and C), genes that are essential for protein folding and are up-regulated upon stress conditions (Srivastava, 2002; Åkerfelt *et al*, 2010). Although genes in HSP pathways are up-regulated in some cancers and have known roles in tumour cell survival (Trachootham *et al*, 2009), thus far this gene family has received little attention in the context of CLL. Consistent with

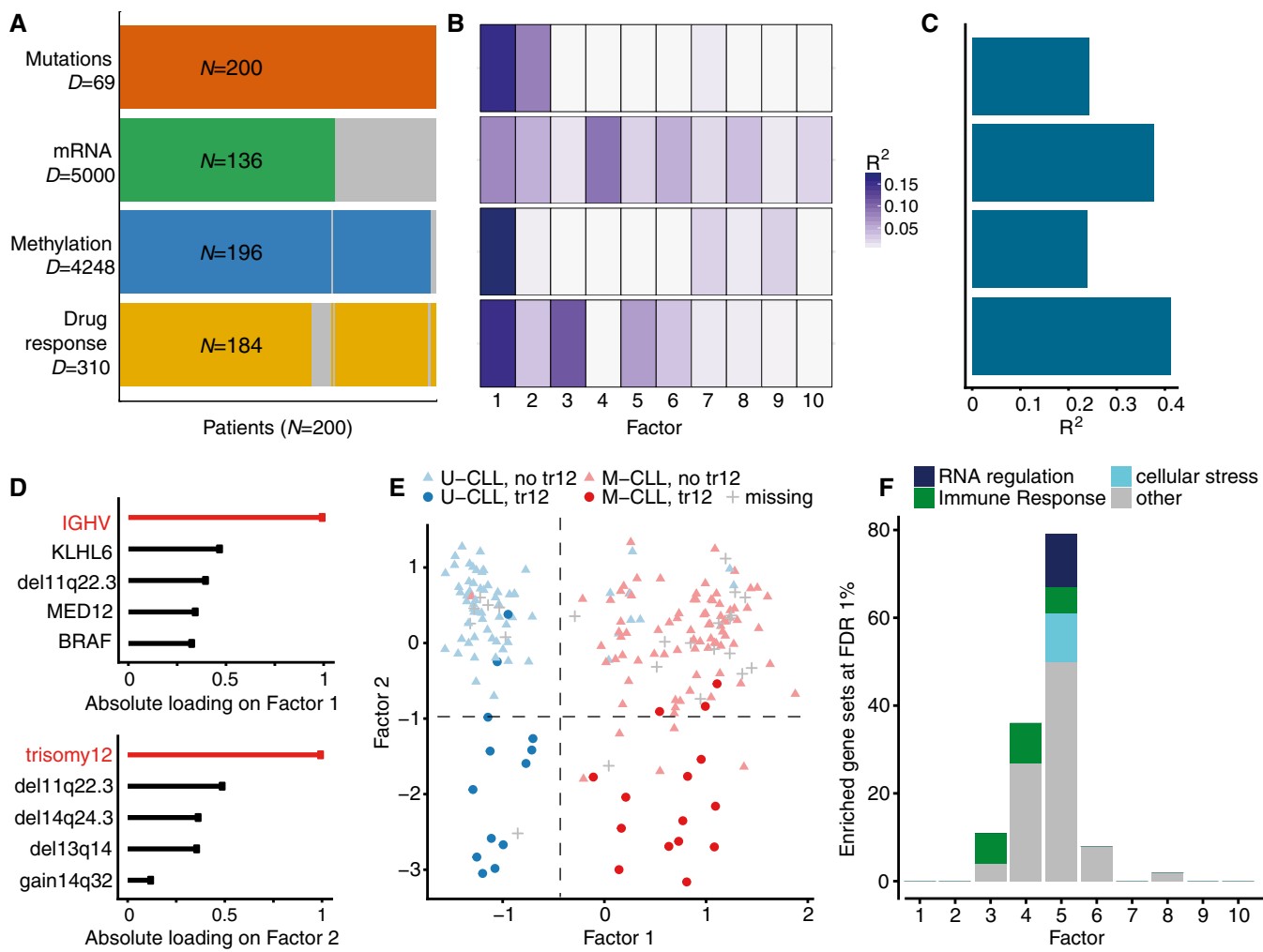

**Figure 2.  Application of MOFA to a study of chronic lymphocytic leukaemia.**

A    Study overview and data types. Data modalities are shown in different rows (*D* = number of features) and samples (*N*) in columns, with missing samples shown using grey bars.

B, C    (B) Proportion of total variance explained ($R^2$) by individual factors for each assay and (C) cumulative proportion of total variance explained.

D    Absolute loadings of the top features of Factors 1 and 2 in the Mutations data.

E    Visualization of samples using Factors 1 and 2. The colours denote the IGHV status of the tumours; symbol shape and colour tone indicate chromosome 12 trisomy status.

F    Number of enriched Reactome gene sets per factor based on the gene expression data (FDR < 1%). The colours denote categories of related pathways defined as in Appendix Table S2.

this annotation based on the mRNA data, we observed that the drugs with the strongest weights on Factor 5 were associated with response to oxidative stress, such as target reactive oxygen species (ROS), DNA damage response and apoptosis (Fig EV2D and E).

Factor 4 captured 9% of variation in the mRNA data, and gene set enrichment analysis on the mRNA loadings suggested aetiologies related to immune response pathways and T-cell receptor signalling (Fig 2F), likely due to differences in cell type composition between samples: While the samples are comprised mainly of B cells, Factor 4 revealed a possible contamination with other cell types such as T cells and monocytes (Appendix Fig S14). Factor 3 explained 11% of variation in the drug response data capturing differences in the samples' general level of drug sensitivity (Geeleher *et al*, 2016; Appendix Fig S15).

## MOFA identifies outlier samples and accurately imputes missing values

Next, we explored the relationship between inferred factors and clinical annotations, which can be missing, mis-annotated or inaccurate, since they are frequently based on single markers or imperfect surrogates (Westra *et al*, 2011). Since IGHV status is the major biomarker impacting on clinical care, we assessed the consistency between the inferred continuous Factor 1 and this binary marker. For 176 out of 200 patients, the MOFA factor was in agreement with the clinical IGHV status, and MOFA further allowed for classifying 12 patients that lacked clinically measured IGHV status (Fig EV3A and B). Interestingly, MOFA assigned 12 patients to a different group than suggested by their clinical IGHV label. Upon inspection

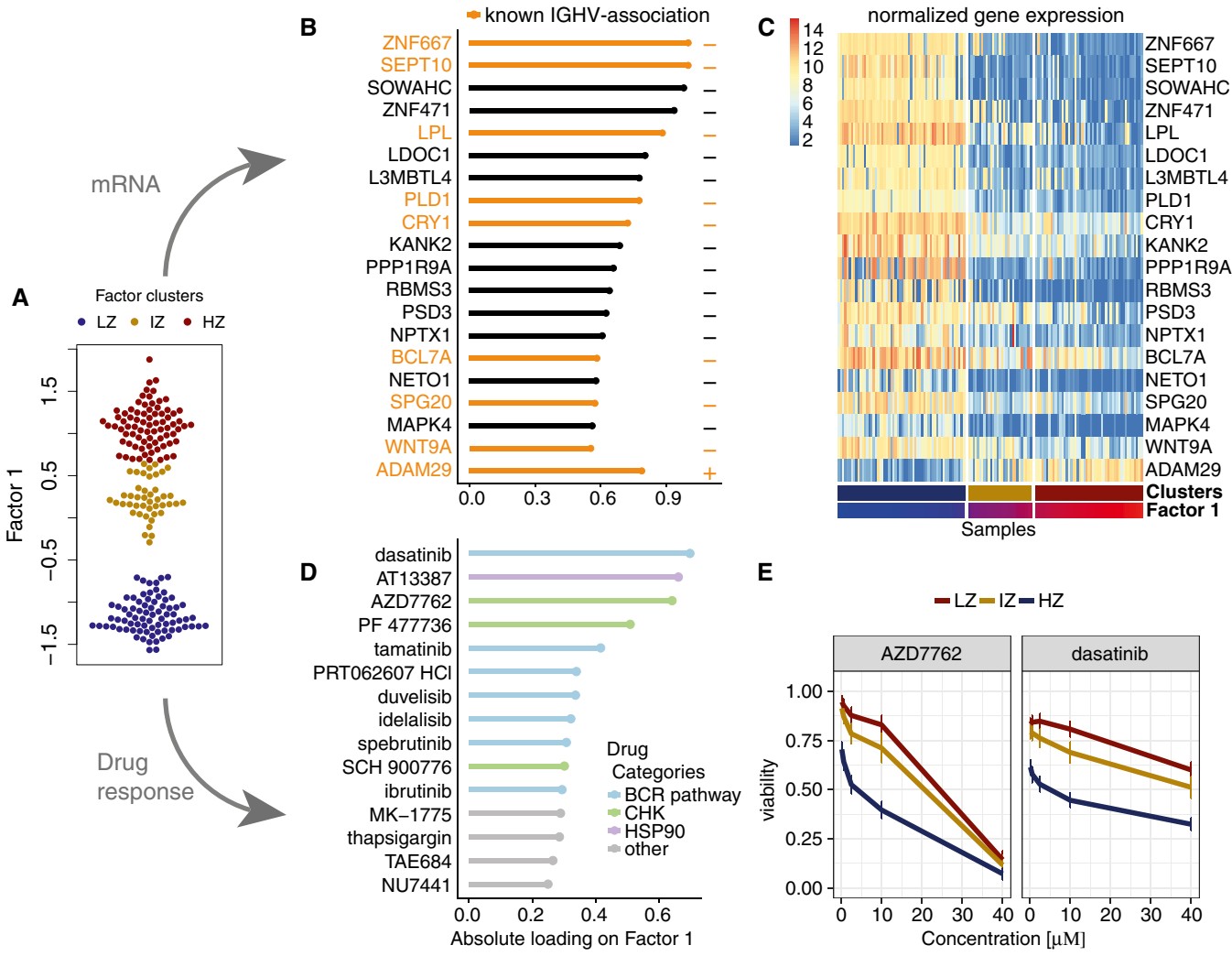

**Figure 3. Characterization of the inferred factor associated with the differentiation state of the cell of origin.**

A  Beeswarm plot with Factor 1 values for each sample with colours corresponding to three groups found by 3-means clustering with low factor values (LZ), intermediate factor values (IZ) and high factor values (HZ).

B  Absolute loadings for the genes with the largest absolute weights in the mRNA data. Plus or minus symbols on the right indicate the sign of the loading. Genes highlighted in orange were previously described as prognostic markers in CLL and associated with IGHV status (Vasconcelos *et al*, 2005; Maloum *et al*, 2009; Trojani *et al*, 2012; Morabito *et al*, 2015; Plesingerova *et al*, 2017).

C  Heatmap of gene expression values for genes with the largest weights as in (B).

D  Absolute loadings of the drugs with the largest weights, annotated by target category.

E  Drug response curves for two of the drugs with top weights, stratified by the clusters as in (A).

of the underlying molecular data, nine of these cases showed intermediate molecular signatures, suggesting that they are borderline cases that are not well captured by the binary classification; the remaining three cases were clearly discordant (Fig EV3C and D). Additional independent drug response assays as well as whole exome sequencing data confirmed that these cases are outliers within their IGHV group (Fig EV3E and F).

As incomplete data is a common problem in studies that combine multiple high-throughput assays, we assessed the ability of MOFA to fill in missing values within assays as well as when entire data modalities are missing for some of the samples. For both imputation tasks, MOFA yielded more accurate predictions than other established imputation strategies, including imputation by feature-wise mean, SoftImpute (Mazumder *et al*, 2010) and a k-nearest neighbour method (Troyanskaya *et al*, 2001; Fig EV4, Appendix Fig S16), and MOFA was more robust than GFA, especially in the case of missing assays (Appendix Fig S17).

## Latent factors inferred by MOFA are predictive of clinical outcomes

Finally, we explored the utility of the latent factors inferred by MOFA as predictors in models of clinical outcomes. Three of the 10 factors identified by MOFA were significantly associated with time to next treatment (Cox regression, Materials and Methods, FDR < 1%, Fig 4A and B): Factor 1, related to the B-cell of origin,

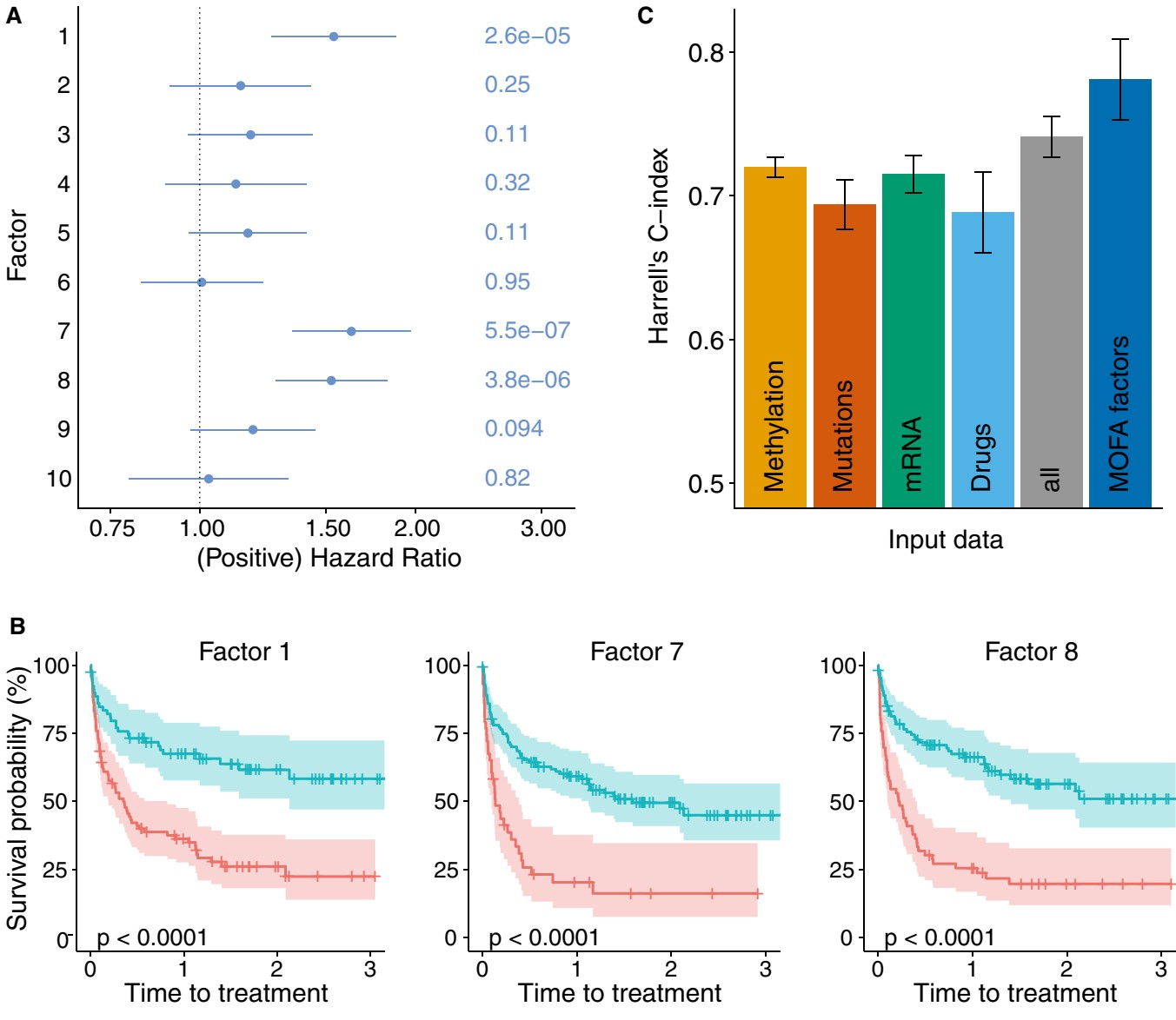

**Figure 4. Relationship between clinical data and latent factors.**

A  Association of MOFA factors to time to next treatment using a univariate Cox regression with *N* = 174 samples (96 of which are uncensored cases) and *P*-values based on the Wald statistic. Error bars denote 95% confidence intervals. Numbers on the right denote *P*-values for each predictor.

B  Kaplan–Meier plots measuring time to next treatment for the individual MOFA factors. The cut-points on each factor were chosen using maximally selected rank statistics (Hothorn & Lausen, 2003), and *P*-values were calculated using a log-rank test on the resulting groups.

C  Prediction accuracy of time to treatment for *N* = 174 patients using multivariate Cox regression trained using the 10 factors derived using MOFA, as well using the first 10 components obtained from PCA applied to the corresponding single data modalities and the full data set (assessed on hold-out data). Shown are average values of Harrell's C-index from fivefold cross-validation. Error bars denote standard error of the mean.

and two Factors, 7 and 8, associated with chemo-immunotherapy treatment prior to sample collection (*P* < 0.01, *t*-test). In particular, Factor 7 captures del17p and TP53 mutations as well as differences in methylation patterns of oncogenes (Garg *et al*, 2014; Fluhr *et al*, 2016; Appendix Fig S18), while Factor 8 is associated with WNT signalling (Appendix Fig S19).

We also assessed the prediction performance when combining the 10 MOFA factors in a multivariate Cox regression model.

Notably, this model yielded higher prediction accuracy than models using components derived from conventional PCA (Fig 4C), individual molecular features (Appendix Fig S20) or MOFA factors derived from only a subset of the available data modalities (Appendix Fig S8B and D; assessed using cross-validation, Materials and Methods). The predictive value of MOFA factors was similar to clinical covariates (such as lymphocyte doubling time) that are used to guide treatment decisions (Appendix Fig S21).

### In an application to single cell data MOFA reveals coordinated changes between the transcriptome and the epigenome along a differentiation trajectory

As multi-omics approaches are also beginning to emerge in single-cell biology (Macaulay *et al*, 2015; Angermueller *et al*, 2016; Guo *et al*, 2017; Clark *et al*, 2018; Colomé-Tatché & Theis, 2018), we investigated the potential of MOFA to disentangle the heterogeneity observed in such studies. We applied MOFA to a data set of 87 mouse embryonic stem cells (mESCs), comprising of 16 cells cultured in "2i" media, which induces a naive pluripotency state, and 71 serum-grown cells, which commits cells to a primed

pluripotency state poised for cellular differentiation (Angermueller *et al*, 2016). All cells were profiled using single-cell methylation and transcriptome sequencing, which provides parallel information of these two molecular layers (Fig 5A). We applied MOFA to disentangle the observed heterogeneity in the transcriptome and the CpG methylation at three different genomic contexts: promoters, CpG islands and enhancers.

MOFA identified three major factors driving cell–cell heterogeneity (minimum explained variance of 2%, Materials and Methods): while Factor 1 is shared across all data modalities (7% variance explained in the RNA data and between 53 and 72% in the methylation data sets), Factors 2 and 3 are active primarily in the RNA data

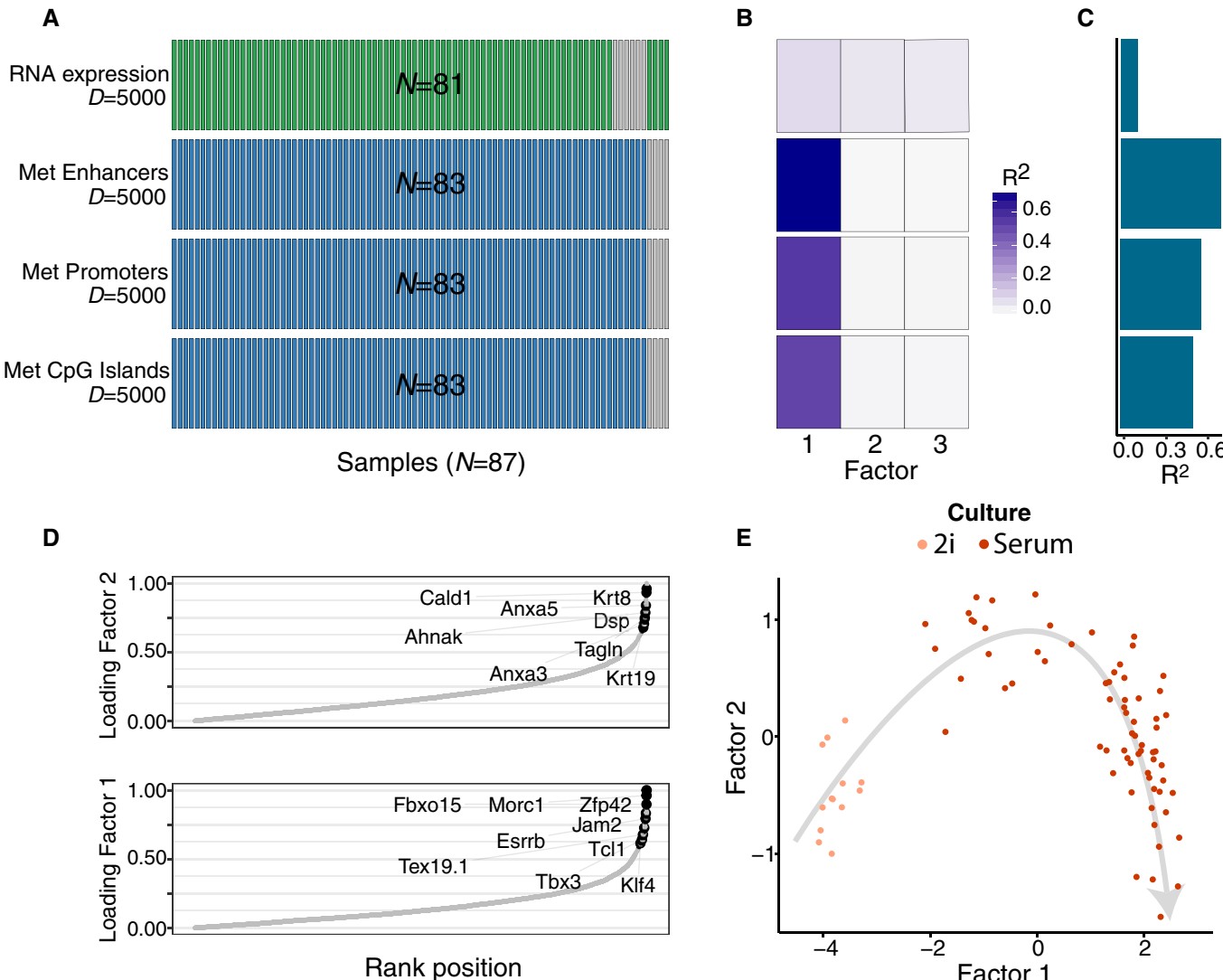

**Figure 5. Application of MOFA to a single-cell multi-omics study.**

A    Study overview and data types. Data modalities are shown in different rows (*D* = number of features) and samples (*N*) in columns, with missing samples shown using grey bars.

B, C  (B) Fraction of the variance explained ($R^2$) by individual factors for each data modality and (C) cumulative proportion of variance explained.

D    Absolute loadings of Factor 1 (bottom) and Factor 2 (top) in the mRNA data. Labelled genes in Factor 1 are known markers of pluripotency (Mohammed *et al*, 2017) and genes labelled in Factor 2 are known differentiation markers (Fuchs, 1988).

E    Scatterplot of Factors 1 and 2. Colours denote culture conditions. The grey arrow illustrates the differentiation trajectory from naive pluripotent cells via primed pluripotent cells to differentiated cells.

(Fig 5B and C). Gene loadings revealed that Factor 1 captured the cells' transition from naïve to primed pluripotent states, pinpointing pluripotency markers such as Rex1/Zpf42, Tbx3, Fbxo15 and Essrb (Mohammed *et al*, 2017; Figs 5D and EV5A). MOFA connected these transcriptomic changes to coordinated changes in the genome-wide DNA methylation rate across all genomic contexts (Fig EV5B), as previously described both *in vitro* (Angermueller *et al*, 2016) and *in vivo* (Auclair *et al*, 2014). Factor 2 captured a second axis of differentiation from the primed pluripotency state to a differentiated state with highest RNA loadings for known differentiation markers such as keratins and annexins (Fuchs, 1988; Figs 5D and EV5C). Finally, Factor 3 captured the cellular detection rate, a known technical covariate associated with cell quality and mRNA content (Finak *et al*, 2015; Appendix Fig S22).

Jointly, Factors 1 and 2 captured the entire differentiation trajectory from naive pluripotent cells via primed pluripotent cells to differentiated cells (Fig 5E), illustrating the importance of learning continuous latent factors rather than discrete sample assignments. Multi-omics clustering algorithms such as SNF (Wang *et al*, 2014) or iCluster (Shen *et al*, 2009; Mo *et al*, 2013) were only capable of distinguishing cellular subpopulations, but not of recovering continuous processes such as cell differentiation (Appendix Fig S23).

## Discussion

Multi-Omics Factor Analysis (MOFA) is an unsupervised method for decomposing the sources of heterogeneity in multi-omics data sets. We applied MOFA to high-dimensional and incomplete multi-omics profiles collected from patient-derived tumour samples and to a single-cell study of mESCs.

First, in the CLL study, we demonstrated that our method is able to identify major drivers of variation in a clinically and biologically heterogeneous disease. Most notably, our model identified previously known clinical markers as well as novel putative molecular drivers of heterogeneity, some of which were predictive of clinical outcome. Additionally, since MOFA factors capture variations of multiple features and data modalities, inferred factors can help to mitigate assay noise, thereby increasing the sensitivity for identifying molecular signatures compared to using individual features or assays. Our results also demonstrate that MOFA can leverage information from multiple omics layers to accurately impute missing values from sparse profiling data sets and guide the detection of outliers, e.g. due to mislabelled samples or sample swaps.

In a second application, we used MOFA for the analysis of single-cell multi-omics data. This use case illustrates the advantage of learning continuous factors, rather than discrete groups, enabling MOFA to recover a differentiation trajectory by combining information from two sparsely profiled molecular layers.

While applications of factor models for integrating different data types were reported previously (Lanckriet *et al*, 2004; Shen *et al*, 2009; Akavia *et al*, 2010; Mo *et al*, 2013), MOFA provides unique features (Materials and Methods, Appendix Table S3) that enable the interpretable reconstruction of the underlying factors and accommodating different data types as well as different patterns of missing data. MOFA is available as open-source software and includes semi-automated analysis pipelines allowing for in-depth characterizations of inferred factors. Taken together, this will foster the accessibility of interpretable factor models for a wide range of multi-omics studies.

Although we have addressed important challenges for multi-omics applications, MOFA is not free of limitations. The model is linear, which means that it can miss strongly non-linear relationships between features within and across assays (Buettner & Theis, 2012). Non-linear extensions of MOFA may address this, although, as with any models in high-dimensional spaces, there will be trade-offs between model complexity, computational efficiency and interpretability (preprint: Damianou *et al*, 2016). A related area of work is to incorporate prior information on the relationships between individual features. For example, future extensions could make use of pathway databases within each omic type (Buettner *et al*, 2017) or priors that reflect relationships given by the "dogma of molecular biology". In addition, new likelihoods and noise models could expand the value of MOFA in data sets with specific statistical properties that hamper the application of traditional statistical methods, including zero-inflated data (i.e. scRNA-Seq; Pierson & Yau, 2015) or binomial distributed data (i.e. splicing events; Huang & Sanguinetti, 2017). Finally, while here we focus our attention on the point estimates of inferred factors, future extensions could attempt a more comprehensive Bayesian treatment that propagates evidence strength and estimation uncertainties to diagnostics and downstream analyses.

## Materials and Methods

### Multi-Omics Factor Analysis model

Starting from $M$ data matrices $\mathbf{Y}^1,..,\mathbf{Y}^M$ of dimensions $N \times D_m$, where $N$ is the number of samples and $D_m$ the number of features in data matrix $m$, MOFA decomposes these matrices as

$$\mathbf{Y^m} = \mathbf{ZW^{mT}} + \boldsymbol{\varepsilon^m} \quad m = 1,\ldots,M. \tag{1}$$

Here, $\mathbf{Z}$ denotes the factor matrix (common for all data matrices) and $\mathbf{W}^m$ denotes the weight matrices for each data matrix $m$ (also referred to as view $m$ in the following). $\boldsymbol{\varepsilon^m}$ denotes the view-specific residual noise term, with its form depending on the specifics of the data type (see Noise model).

The model is formulated in a probabilistic Bayesian framework, where we place prior distributions on all unobserved variables of the model (see plate diagram in Appendix Fig S24), i.e. the factors $\mathbf{Z}$, the weight matrices $\mathbf{W}^m$ and the parameters of the residual noise term. In particular, we use a standard normal prior for the factors $\mathbf{Z}$ and employ sparsity priors for the weight matrices (see next section).

### Model regularization

An appropriate regularization of the weight matrices is essential for the model's ability to disentangle variation across data sets and to yield interpretable factors. MOFA uses a two-level regularization: the first level encourages view- and factor-wise sparsity, thereby allowing to directly identify which factor is active in which view. The second level encourages feature-wise sparsity, thereby typically resulting in a small number of features with active weights. To

encode these sparsity levels, we combine an Automatic Relevance Determination (ARD) prior for the first type of the sparsity with a spike-and-slab prior for the second. For amenable inference, we model the spike-and-slab prior by parameterizing the weights as a product of a Bernoulli distributed random variable and a normally distributed random variable: $W = S\widehat{W}$, where $s_{dk}^m \sim \text{Ber}(\theta_k^m)$ and $\widehat{W}_{dk}^m \sim N(0, 1/\alpha_k^m)$. To automatically learn the appropriate level of regularization for each factor and view, we use uninformative conjugate prior on $\alpha_k^m$, which controls the strength of factor $k$ in view $m$, and on $\theta_k^m$, which determines the feature-wise sparsity level of factor $k$ in view $m$ (see Appendix Supplementary Methods, Section 2 for details).

### Noise model

MOFA supports the combination of different noise models to integrate diverse data types, including continuous, binary and count data. A standard noise model for continuous data is the Gaussian noise model assuming iid heteroscedastic residuals $\boldsymbol{\varepsilon}^{\mathbf{m}}$, i.e. $\varepsilon_{nd}^m \sim N(0, 1/\tau_d^m)$, with Gamma prior on the precision parameters $\tau_d^m$. MOFA further supports noise models for binary and count data that are not appropriately modelled using a Gaussian likelihood. In the current version, MOFA models count data using a Poisson model and binary data by using a Bernoulli model. Here, the model likelihood is given by $y_{nd}^m \sim \text{Poi}\big(\lambda(Z_{n:}w_{d:}^T)\big)$ and $y_{nd}^m \sim \text{Ber}\big(\sigma(Z_{n:}w_{d:}^T)\big)$, respectively, where $\lambda(x) = \log(1 + e^x)$ and $\sigma$ denotes the logistic function $\sigma(x) = (1 + e^{-x})^{-1}$.

### Parameter inference

For scalability, we make use of a variational Bayesian framework, which is essentially a mean field approximation for approximate inference (Blei *et al*, 2017). The key idea is to approximate the intractable posterior distribution using a simpler class of distributions by minimizing the Kullback–Leibler divergence to the exact posterior, or equivalently maximizing the evidence lower bound (ELBO). Convergence of the algorithm can be monitored based on the ELBO. An overview of variational inference and details on the specific implementation for MOFA can be found in Appendix Supplementary Methods, Section 3. To enable an efficient inference for non-Gaussian likelihoods, we employ variational lower bounds on the likelihood (Jaakkola & Jordan, 2000; Seeger & Bouchard, 2012; see Appendix Supplementary Methods, Section 4).

### Model training and selection

An important part of the training is the determination of the number of factors. Factors are automatically inactivated by the ARD prior of the model as described in Model regularization. In practice, factors are pruned during training using a minimum fraction of variance explained threshold that needs to be specified by the user. Alternatively, the user can fix the number of factors and the minimum variance criterion is ignored. In the analyses presented, we initialized the models with $K = 25$ factors and they were pruned during training using a threshold of variance explained of 2%. For details on the implementation as well as practical considerations for training and choice of the threshold parameter, refer to Appendix Supplementary Methods, Section 5.

While the inferred factors are robust under different initializations (e.g. Appendix Fig S6C and D), the optimization landscape is non-convex, and hence, the algorithm is not guaranteed to converge to a global optimum. Results presented here are based on 10–25 random restarts, selecting the model with the highest ELBO (e.g. Appendix Fig S6B).

### Downstream analysis for factor interpretation and annotation

As part of MOFA, we provide the R package *MOFAtools,* which provides a semi-automated pipeline for the characterization and interpretation of the latent factors. In all downstream analyses, we use the expectations of the model variables under the posterior distributions inferred by the variational framework.

The first step, after a model has been trained, is to disentangle the variation explained by each factor in each view. To this end, we compute the fraction of the variance explained ($R^2$) by factor $k$ in view $m$ as

$$R_{m,k}^2 = 1 - \left(\sum_{n,d} y_{nd}^m - z_{nk}w_{kd}^m - \mu_d^m\right)^2 \Big/ \left(\sum_{n,d} y_{nd}^m - \mu_d^m\right)^2$$

as well as the fraction of variance explained per view taking into account all factors

$$R_m^2 = 1 - \left(\sum_{n,d} y_{nd}^m - \sum_k z_{nk}w_{kd}^m - \mu_d^m\right)^2 \Big/ \left(\sum_{n,d} y_{nd}^m - \mu_d^m\right)^2$$

Here, $\mu_d^m$ denotes the feature-wise mean. Subsequently, each factor is characterized by three complementary analyses:

1   *Ordination of the samples in factor space:* Visualize a low-dimensional representation of the main drivers of sample heterogeneity.
2   *Inspection of top features with largest weight:* The loadings can give insights into the biological process underlying the heterogeneity captured by a latent factor. Due to scale differences between assays, the weights of different views are not directly comparable. For simplicity, we scale each weight vector by its absolute value.
3   *Feature set enrichment analysis:* Combine the signal from functionally related sets of features (e.g. gene sets) to derive a feature set-based annotation. By default, we use a parametric *t*-test comparing the means of the foreground set (the weights of features that belong to a set $G$) and the background set (the weights of features that do not belong to the set $G$), similar to the approach described in Frost *et al* (2015).

### Relationship to existing methods

MOFA builds upon the statistical framework of group Factor Analysis (Virtanen *et al*, 2012; Khan *et al*, 2014; Klami *et al*, 2015; Bunte *et al*, 2016; Zhao *et al*, 2016; Leppäaho & Kaski, 2017) and is in part also related to the iCluster methods (Shen *et al*, 2009; Mo *et al*, 2013) as shown in Appendix Table S3. Here, we describe these connections in further detail.

#### iCluster
In contrast to MOFA, iCluster uses in each view the same extent of regularization for all factors, which may be sufficient for the purpose of clustering (the primary application of iCluster); however,

it results in a reduced ability for distinguishing factors that drive variation in distinct subsets of views (Appendix Fig S5). Additionally, unlike MOFA and GFA, iCluster does not handle missing values and is computationally demanding (Fig EV1), as it requires re-fitting the model for a large range of different penalty parameters and choices of the model dimension.

### Group Factor Analysis

While the underlying model of MOFA is closely connected to the most recent GFA implementation (Leppäaho & Kaski, 2017), GFA is restricted to Gaussian observation noise. In terms of the algorithmic implementation, MOFA uses an additional "burn-in period" during training during which the sparsity constraints are deactivated to avoid early splitting of factors and actively drops factors below a predefined variance threshold (see Model training and selection). In contrast, GFA directly uses sparsity constraints throughout training and also maintains factors that have near-zero relevance. In terms of inference, MOFA is implemented using a variational approximate Bayesian inference, whereas GFA is based on a Gibbs sampler. In terms of computational scalability (Fig EV1), both methods are linear in the model's parameters, although GFA is computationally more expensive in absolute terms. This difference is particularly pronounced for data sets with missing data. This, together with the inability to deactivate factors during inference (Appendix Fig S4), renders GFA considerably slower in applications to real data.

### Details on the simulation studies

#### Model validation

To validate MOFA, we simulated data from the generative model for a varying number of views ($M = 1,3,\ldots,21$), features ($D = 100,500,\ldots,10,000$), factors ($K = 5,10,\ldots,60$), missing values (from 0 to 90%) as well as for non-Gaussian likelihoods (Poisson, Bernoulli; see Appendix Table S1 for simulation parameters). We assessed the ability of MOFA to recover the true simulated number of factors in the different settings, where we considered 10 repeat experiments for every configuration. All trials were started with a high number of factors ($K = 100$), and inactive factors were pruned as described in Model training and selection.

#### Model comparison

To compare MOFA with to GFA, we simulated data from the underlying generative model with $K_{\text{true}} = 10$ factors, $M = 3$ views, $N = 100$ samples, $D = 5,000$ features each and 5% missing values (missing at random). For each of the three views, we used a different likelihood model: continuous data were simulated with a Gaussian distribution, binary data with a Bernoulli distribution and count data with a Poisson distribution. Except for the non-Gaussian likelihood extension, both methods share the same underlying generative model, thus allowing for a meaningful comparison. We fit ten realizations of the MOFA and GFA models with $K_{\text{initial}} = 20$ factors and let the method determine the most likely number factors. To assess scalability, we considered the same base parameter settings, varying one of the simulation parameters at a time (number of factors $K$, number of features $D$, number of samples $N$ and number of views $M$, all Gaussian). To assess the ability to reconstruct factor activity patterns, we

simulated data from the generative model for $K_{\text{true}} = 10$ and $K_{\text{true}} = 15$ factors ($M$, $N$, $D$ as before, no missing values, only Gaussian views), where factors were set to either active or inactive in a specific view by sampling the parameter $\alpha_k^m$ from $\{1,10^3\}$. Appendix Table S1 shows in more detail the simulation parameters used in each setting.

### Details on the CLL analysis

#### Data processing

The data were taken from (Dietrich *et al*, 2018), where details on the data generation and processing can be found. Briefly, this data set consists of somatic mutations (combination of targeted and whole exome sequencing), RNA expression (RNA-Seq), DNA methylation (Illumina arrays) and *ex vivo* drug response screens (ATP-based CellTiter-Glo assay). For the training of MOFA, we included 62 drug response measurements (excluding NSC 74859 and bortezomib due to bad quality) at five concentrations each ($D = 310$) with a threshold at 1.1 to remove outliers. Mutations were considered if present in at least three samples ($D = 69$). Low counts from RNA-Seq data were filtered out and the data were normalized using the *estimateSizeFactors* and *varianceStabilizingTransformation* function of DESeq2 (Love *et al*, 2014). For training, we considered the top $D = 5,000$ most variable mRNAs after exclusion of genes from the Y chromosome. Methylation data were transformed to M-values, and we extracted the top 1% most variable CpG sites excluding sex chromosomes ($D = 4,248$). We included patients diagnosed with CLL and having data in at least two views into the MOFA model leading to a total of $N = 200$ samples.

#### Model training and selection

We trained MOFA using 25 random initializations with a variance threshold of 2% and selected the model with the best fit for downstream analysis (see Model training and selection).

#### Gene set enrichment analysis

Gene set enrichment analysis was performed based on Reactome gene sets (Fabregat *et al*, 2015) as described above. Resulting *P*-values were adjusted for multiple testing for each factor using the Benjamini–Hochberg procedure (Benjamini & Hochberg, 1995). Significant enrichments were at a false discovery rate of 1%.

#### Imputation

To compare imputation performance, we trained MOFA on the subset of samples with all measurements ($N = 121$) and masked at random either single values or all measurements for randomly selected samples in the drug response. After model training, the masked values were imputed directly from the model equation (1) and the accuracy was assessed in terms of mean squared error on the true (masked) values. For both settings, we fixed the number of factors in MOFA to $K = 10$. To investigate the dependence on $K$ for imputation and to compare MOFA to GFA, we re-ran the same masking experiments varying $K = 1,\ldots,20$ (Appendix Fig S17).

### Survival analysis

Associations between the inferred factors and clinical covariates were assessed using the patients' time to next treatment as

response variable in a Cox model ($N$ = 174 samples with treatment information, 96 of which are uncensored cases). For univariate association tests (as shown in Fig 4A, Appendix Fig S21), we scaled all predictors to ensure comparability of the hazard ratios and we rotated factors, which are rotational invariant, such that their hazard ratio is greater or equal to 1. To investigate the predictive power of different data sets, we used a multivariate Cox model and compared Harrell's C-index of predictions in a stratified fivefold cross-validation scheme. As predictors, we included the top 10 principal components calculated on the data for each single view, a concatenated data set ("all") as well as the 10 MOFA factors. Missing values in a view were set to the feature-wise mean. In a second set of models, we used the complete set of all features in a view with a ridge penalty in the Cox model as implemented in the R package *glmnet*. For the Kaplan–Meier plots, an optimal cut-point on each factor was determined to define the two groups using the maximally selected rank statistics as implemented in the R package *survminer* with $P$-values based on a log-rank test between the resulting groups.

**Details on the scMT analysis**

The data were obtained from Angermueller *et al* (2016), where details on the data generation and pre-processing can be found. Briefly for each CpG site, we calculated a binary methylation rate from the ratio of methylated read counts to total read counts. RNA expression data were normalized using Lun *et al* (2016). To fit MOFA, we considered the top 5,000 most variable genes with a maximum dropout of 90% and the top 5,000 most variable CpG sites with a minimum coverage of 10% across cells. Model selection was performed as described for the CLL data, and factors were inactivated below a minimum explained variance of 2%. For the clustering analysis using SNF and iCluster, the optimal number of clusters was selected using the BIC criterion.

**Data and software availability**

- The CLL data were obtained from Dietrich *et al* (2018) and are available at the European Genome–Phenome Archive under accession EGAS00001001746 and data tables as R objects can be downloaded from http://pace.embl.de/. The single-cell data were obtained from Angermueller *et al* (2016) and are available in the Gene Expression Omnibus under accession GSE74535. All data used are contained within the MOFA vignettes and can be downloaded as from https://github.com/bioFAM/MOFA.
- An open-source implementation of MOFA in R and Python is available from https://github.com/bioFAM/MOFA. Code to reproduce all the analyses presented is available at https://github.com/bioFAM/MOFA_analysis.

**Expanded View** for this article is available online.

## Acknowledgements

We thank everybody involved in the generation and analysis of the original CLL study for sharing their data and analysis ahead of publication, especially M. Oleś for providing the associated data package and to J. Lu, J. Hüllein and A. Mock for discussions on CLL biology. The work was supported by the European Union (Horizon 2020 project SOUND) and project PanCanRisk.

## Author contributions

RA and BV contributed equally and are listed alphabetically. FB, DA and OS conceived the model. RA, DA and BV implemented the model. TZ, SD and WH designed the CLL study and generated the data. RA and BV performed the analysis. RA, BV, DA, TZ, SD, WH, OS, FB and JCM interpreted the results. RA, BV, OS, WH and FB conceived the project. RA, BV, OS, FB and WH wrote the manuscript. OS, WH, FB and JCM supervised the project.

## Conflict of interest

The authors declare that they have no conflict of interest.

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
