## [Review Process File · Molecular Systems Biology]

Multi-Omics Factor Analysis—a framework for unsupervised integration of multi-omics data sets

Ricard Argelaguet, Britta Velten, Damien Arnol, Sascha Dietrich, Thorsten Zenz, John C. Marioni, Florian Buettner, Wolfgang Huber, Oliver Stegle

Review timeline:

Submission date:	27 th November 2017
Editorial Decision:	10 th January 2018
Revision received:	9 th April 2018
Editorial Decision:	16 th May 2018
Revision received:	28 th May 2018
Accepted:	29 th May 2018

Editor: Maria Polychronidou

Transaction Report:

1st Editorial Decision

10th January 2018

Thank you again for submitting your work to Molecular Systems Biology. We have now heard back from the three referees who agreed to evaluate your study. As you will see below, the reviewers appreciate that the presented approach seems potentially useful. They raise however a series of concerns, which we would ask you to address in a revision of the manuscript.

The reviewers' recommendations are rather clear so I think that there is no need to repeat all the points listed below. Reviewer #1 refers to the need to include comparisons of the method to existing approaches. Importantly, all reviewers point out that the biological insights from the CLL analysis remain rather limited. As reviewer #3 recommends, additional analyses (not necessarily experimental), demonstrating the potential of the method to reveal new biological insights would significantly enhance the impact of the study. Of course all other issues raised by the referees would need to be thoroughly addressed.

REFERE REPORTS

Reviewer #1:

With the increasing availability of multi-omics datasets, the development of statistical methods to disentangle specific- and shared-components of variation in such datasets has numerous applications and will undoubtedly help to expose novel biology. The authors present Multi-Omics Factor Analysis (MOFA) which applies Group Factor Analysis to identify hidden factors that capture biological and technical sources of variability. The model builds on group factor analysis with the additional optimizations of (a) fast inference based on variational approximation, (b) inference of sparse solutions by inducing group-wise and feature-wise sparsity, (c) handling missing values and

(d) integration of different data likelihood models to integrate non-Gaussian data. The manuscript is well written, the method is evaluated with extensive simulations and further highlights the application of MOFA to CLL to identify and enhance clinical markers. However, the method and its application to CLL have several challenges that need to be further explored and dampen its impact. A major concern is that the method is not contrasted in practice to other group factor analysis methods that are closely related to MOFA (listed in Table 1 of Supplemental Methods), neither in the simulation study nor the real data analysis. While the iCluster method is widely used, it would be important to minimally see a comparison to the Hore et al (2016) method, which is the method most closely related to MOFA.

- 1) The definition of "views" is not well articulated.
- 2) While the extension of the method to non-Gaussian data performs reasonably well, it assumes the same distribution across data modalities. This is almost never the case in practice when performing multi-omics analyses, e.g. when modelling both binary SNP data and count gene expression data. It is thus not exactly clear what the advantage of the extension is.
- 3) In the analysis of the chronic lymphocytic leukemia data, not much information is provided about the choice of K in the three-mean clustering of samples based on factor 1 (Figure 3A). Was there a formal test about the goodness-of-fit of a three- versus e.g. two-mean clustering? The IZ and HZ clusters do not show a large difference in gene expression values (Figure 3C) or Drug response (Figure 3E).
- 4) Even though factor 8 was significantly associated with time to next treatment (Figure 5A) and it explains some proportion of the gene expression variability, the factor shows no enrichment based on the GSEA (Figure 2E). Moreover, there is not much information / knowledge about what this factor is correlated to besides the one line in the text 'Factor 8 is associated with differences in gene expression between pre-treated and untreated patients'. Can you elaborate more about Factor 8? For example provide a plot similar to Sup Figure 15?
- 5) The CLL variable is time to next treatment. It is very likely that individuals currently or proximally treated will have important changes in gene expression and that simply knowing if a person is being actively treated or not plus a few other clinical variables could be very helpful in predicting the timing of the next treatment. Particularly in the context of how CLL and its therapy may be timed in practice. This is highlighted in the observation that the response, next treatment timing, is best predicted by factors correlated to whether the patient was chemo-immunotherapy treated before collected. The relationship of the predictors to clinical variables such as these is not adequately explored and ultimately it is difficult to assess whether the molecular approach is better than an alternative approach that uses a combination of clinical phenotypes (like cell counts or further accounts for accelerated treatment for increased disease severity).
- 6) Is the data for the CLL example downloadable with MOFA?

Minor:

- 1) Figure S2c appears to show inflation of number of inferred factors when using binary data. Figure S3c is also inflated for Gaussian data. Can the authors comment on this?
- 2) Page 3, second paragraph, add citations for the iCluster method after 'Comparison of MOFA with the factor model that underlies the widely-used iCluster method (Shen et al 2009)' and other related approaches after 'We contrast MOFA to related approaches in Methods and Supp. Table 1.'
- 3) Page 3, second paragraph: 'We contrast MOFA to related approaches in Methods and Supp. Table 1'. The correct reference for this is "Supplemental Methods" and "Table 1 of Supplemental Methods" (not Supplemental Table 1 which refers to simulation analysis parameters).
- 4) All figure reference in "Supplemental Methods" are off by one number. In section 2.1, it's Figure S1 that contains results for recovery of the number of factors and not Figure S2. In section 2.2, it's Figure S2 and Figure S3 that contain results for non-Gaussian data and not Figure S3 and Figure S4 etc.
- 5) Figure S5b doesn't appear to show R2 and colors on the same scale.
- 6) For Figure S7b can the authors comment on the trend for Factor 1?

Reviewer #2:

Argelaguet et al. present algorithms and code to implement group factor analysis for multi-omics data, which they term MOFA. They demonstrate the utility of MOFA on a public multi-omics data cohort, derived from leukaemia clinical samples.

Different variants of factor analysis, canonical correlation analysis and principal components analysis are widely used in systems biology studies. However, the authors have addressed some of the computational issues involved in applying group factor analysis to multi-omics data, including: code for fast inference, inference of sparse solutions, handling missing values, and likelihood models for integrating Boolean and count data.

The manuscript addresses issues encountered in applying group factor analysis to multi-omics data. The use of variational Bayes for fast inference, inference of sparse solutions and the handling of missing values have all been previously described. The use of Poisson and Negative Binomial likelihoods for factor analysis has been previously described. The analysis of CLL data is interesting and demonstrates the power of the approach, but these results are not followed up on in any way.

The code is well written and well documented, and likely to be useful for bioinformatics practitioners with a strong statistical background. The manuscript is clearly written, the supplement provides an excellent primer on factor analysis for multi-omics, useful for readers with a strong mathematics background.

Major:

Although the code appears to be a careful implementation of group factor analysis for multi-omics data, and would likely be useful, it is not clear what the methodological advance of the study is. By their own admission, Table 1 Supp. 1.5, the advances of the study (as described in the introduction) have all been developed previously. Although not listed in that table, there are other implementations of Poisson and NB likelihoods too.

The application of MOFA to CLL data is interesting, and very carefully executed. But, there has been no follow-up studies performed. Thus, it is difficult to say that this manuscript represents a vertical advance in the field of CLL research.

Ultimately, the main contribution of the manuscript appears to be the code and the tutorial, which does seem very useful. I would ask the authors to consider either developing the CLL analysis through additional molecular assays, and refocus the paper on those results. Or, more clearly delineate how MOFA is a methodological advance over existing GFA approaches.

Minor:

The title "Multi-Omics factor analysis disentangles heterogeneity in blood cancer", suggests that this is a manuscript about blood cancers. However, the only result for blood cancer I can find in the abstract is: "... [we] discovered previously underappreciated drivers of variation, such as response to oxidative stress.", which is somewhat of a modest result from which no conclusions are drawn. The manuscript itself reads like a methods paper, and in my opinion would make a strong software note. But, the title should reflect the ultimate direction the authors choose for the manuscript.

Reviewer #3:

The manuscript presents an elegant Bayesian model "MOFA" based on group factor analysis to infer factors explaining variation across multiple omics datasets. There is a strong need for computational methods that allow integration of multiple data modalities such as expression, mutational and clinical data collected from the same system and this paper presents a principled methodology that provides interpretable factors for studying heterogeneity across samples from integration of datasets. The authors design a variational inference procedure that includes nice mean-field approximation for non-Gaussian data with two instructive examples for Bernoulli and Poisson data.

The factor space can be used for analysis and visualization, identification of outliers and imputing missing data. The authors show the performance on both simulated and CLL data. They identify the drivers of variation in CLL and molecular signatures based on integration of features and data modalities. They show that the inferred factors explain differences in expression, mutations, methylation or treatment outcome, etc.

This method is a very useful resource and tool for studying heterogeneity in a global scale through combination of matched data types. The model has the advantage to handle large amounts of clinical data, which is common.

I would very much like to see it published at MSB, but feel that some more analysis and interpretation is still needed, to make this into a better paper. See major comments below.

Major comments

- Based on inferred factors on CLL patient data, the authors show MOFA's Ability of detecting outlier samples as well as imputing missing data in drug response. The simulation experiments showing the performance in imputing missing drug response data was great. However, I wasn't fully convinced with the outlier detection analysis. If the authors believe that some patients were mislabeled based on IGHV status (and were not borderline cases), they should examine what other markers or mechanisms are better surrogates of CLL biology and validate that experimentally.
- Why are factors 3 and 4 not discussed in the paper? Factor 3 is also active in the drug response view, what does it represent?
- Supplementary Figure 11 shows Factor 1 when one view is masked, what are the biological signals that are missed with masking each view? Can you extend this to other factors?
- Also, how would the results look like if only one view was used (and the rest were masked)? Which views were most informative of clinical outcome or CLL markers?
- Overall, while the computational methodology is strong, the paper does not show MOFA's ability in inferring deep biological insight. Therefore, I suggest either performing deeper analysis and biological experiments or applying this method to a second system such as ENCODE datasets to show broader application and insight.

Minor points

- Supplementary methods: Section 1.1 refers to Figure S1 for plate model, should be Figure S17.
- Figure 2E: Were no gene sets enriched for factor 1,2?
- Figure 2F not cited in main text or explained. Some subfigures are not cited in order.
- Figure 4b: Was the gene set enrichment done between the two factor clusters (marked in Figure 4a)? The samples don't show discrete clusters for factor 5 unlike factor 1. What is the significance of clustering factor 5 into two clusters?
- Page 5 second paragraph from bottom: reference should be to Fig 2f not Fig 2d.
- Fig S9: Would be great to show the sample number breakdown by views, to see if factor clusters are associated with (or biased to) views.

We thank the reviewers for their time and effort spent reviewing our work, and for their insightful comments.

Reviewer #1:

With the increasing availability of multi-omics datasets, the development of statistical methods to disentangle specific- and shared-components of variation in such datasets has numerous applications and will undoubtedly help to expose novel biology. The authors present Multi-Omics Factor Analysis (MOFA) which applies Group Factor Analysis to identify hidden factors that capture biological and technical sources of variability. The model builds on group factor analysis with the additional optimizations of (a) fast inference based on variational approximation, (b) inference of sparse solutions by inducing group-wise and feature-wise sparsity, (c) handling missing values and (d) integration of different data likelihood models to integrate non-Gaussian data. The manuscript is well written, the method is evaluated with extensive simulations and further highlights the application of MOFA to CLL to identify and enhance clinical markers. However, the method and its application to CLL have several challenges that need to be further explored and dampen its impact.

A major concern is that the method is not contrasted in practice to other group factor analysis methods that are closely related to MOFA (listed in Table 1 of Supplemental Methods), neither in the simulation study nor the real data analysis. While the iCluster method is widely used, it would be important to minimally see a comparison to the **Hore et al (2016) method**, which is the method most closely related to MOFA.

We agree with the reviewer that an appropriate comparison and benchmark of new methods is essential.

In response, we have now extended the discussion of previous methods and provide additional data on how they relate to MOFA. Concerning the specific method proposed by Hore et al. (2016), we would like to point out that a direct comparison is not feasible. While based on a related mathematical formulation, this method has different aims and applications. Briefly, the Hore et al. model performs a tensor decomposition that assumes as input a 3D array containing genomic data assayed for various tissues and individuals. The dimensions of this tensor are (1) the individuals, (2) the tissues and (3) genes. Importantly, this decomposition makes strong assumptions about the structure of the datasets, namely that for each tissue (corresponding to a view in our model) and individual (sample) the same set of genes (features) is assayed. While this assumption is fulfilled and appropriate for the integration of multiple expression datasets as considered in Hore et al., it does not hold in the multi-omics setting, where different features are measured by each assay (e.g. genome, transcriptome, DNA methylation, drug responses). It is this latter setup that MOFA is designed for.

More pertinent is the comparison of MOFA with the Group Factor Analysis (GFA) by Leppäaho et al. (2017), for which an implementation exists (R package GFA). We now compare MOFA to GFA using simulated and real data (See the new section *Model validation and comparison on simulated data* (page 4), **Figure EV1**, **Appendix Figure S4-S5**, **S9** and

Methods). Briefly, although the generative models that underlie MOFA and GFA are related, our method and its implementation offers three main advantages:

- **MOFA supports more versatile noise models:** MOFA allows for different likelihood models and combinations thereof, which is critical for integrating multi-omics assays with different data modalities. We illustrate this by considering the combination of Gaussian and non-Gaussian likelihoods in the CLL study and in an additional single-cell data set. In contrast, existing group factor analysis methods (including the recent Leppäaho et al. (2017) GFA implementation) assume Gaussian noise.
- **MOFA is computationally more efficient:** While both the Leppäaho et al. GFA implementation and MOFA scale linearly in the model dimensions (views, samples, factors and features), MOFA is implemented using a computationally more efficient variational approximation, compared to the Gibbs sampler used by Leppäaho et al. This leads to major speedups in relevant applications (e.g., for the CLL data a MOFA model can be trained in 45 minutes compared to 33.6 hours with GFA using the same number of initial factors¹). The largest gains in computational efficiency are observed for datasets with missing values (See discussion in **Methods** ‘*Relationship to existing methods*’ section). Increased computational speed facilitates the analysis of larger datasets, enables interactive and explorative use and renders resampling schemes much more practical (e.g. to enable the model selection across multiple restarts as in **Appendix Figure S2-3**, or to enable bootstrap to assess robustness as in **Appendix Figure 7**).
- **Improved selection of model complexity (i.e., number of factors).** MOFA implements a simple yet effective heuristic to determine the dimensionality of the latent space by means of the stepwise deactivation of unused factors, based on a minimum variance criterion (see **Methods** for details). We find that this approach is superior to existing approaches, including the criterion used by Leppäaho et al., and in particular leads to the inference of less spurious factors.

In addition to these core performance parameters, we would like to highlight the improved usability of MOFA, as it comes as a well documented software with a suite of ‘downstream’ analysis methods, including tools for visualisation, automatic annotation of factors, the identification of outliers, clustering and imputation of missing values.

1) The definition of "views" is not well articulated.

We agree that the terminology ‘view’ was not sufficiently well defined in the previous version of the manuscript. We refer to ‘view’ as one of the input data matrices in MOFA, which in the CLL application is equivalent to an omic type. To avoid confusion we now limit the use of the word ‘view’ to the technical description of MOFA in **Methods**, where the term is defined explicitly. In the main text we now explicitly refer to the corresponding omic modality or assay.

¹ Comparison based on 10 runs on a single CPU each, randomly allocated to Intel Xeon Processor E5-2670, E5-2680v3 and Dual AMD EPYC 7601 Processor

2) While the extension of the method to non-Gaussian data performs reasonably well, it assumes the same distribution across data modalities. This is almost never the case in practice when performing multi-omics analyses, e.g. when modelling both binary SNP data and count gene expression data. It is thus not exactly clear what the advantage of the extension is.

We thank the reviewer for highlighting the importance of combining views with different likelihood models. MOFA is not limited to a single (possibly non-Gaussian) likelihood across views and instead can be configured to enable arbitrary combinations of likelihood models. For example, in the CLL application, we employ a Gaussian likelihood for the mRNA, methylation and drug response assays, combined with a Bernoulli likelihood for the mutation assay. We now state this flexibility more clearly in the main text (last paragraph of page 3 and 4). Additionally, we now present a second application of MOFA to a single-cell multiomics dataset, where we model three non-Gaussian views (with Bernoulli likelihood) together with one Gaussian view.

3) In the analysis of the chronic lymphocytic leukemia data, not much information is provided about the choice of K in the three-mean clustering of samples based on factor 1 (Figure 3A). Was there a formal test about the goodness-of-fit of a three- versus e.g. two-mean clustering? The IZ and HZ clusters do not show a large difference in gene expression values (Figure 3C) or Drug response (Figure 3E).

While the Factors inferred by MOFA are inherently continuous, clustering of the samples in the factor space (or subspaces) can be considered during downstream analysis. To assess the choice of number of clusters we used, as suggested, the Bayesian Information Criterion (BIC) for the K -means clustering using the R package ClusterR. The lowest BIC is observed for $K=3$, consistent with our previous choice (See **Appendix Figure S10**).

Regarding the gene expression and drug response profiles of IZ and HZ, it is true that these are relatively similar when compared to the LZ cluster. However, we still find that a considerable subset of the 20 genes and 10 concentration points shown in **Figure 3** exhibit significant differences between the groups (10 genes and 7 concentration points, $P < 0.05$, t-test, see Figure below). Nevertheless, we agree that the differences between IZ and HZ are most prominent for the methylation data. These data most likely drive the observed factor clusters, which are in strong agreement with *methylation clusters* defined by others (**Appendix Figure S11**).

Cluster-wise comparison of the genes and drugs shown in Figure 3.

Boxplots for gene expressions levels (a) and drug response (b) for the top-weighted genes and drugs shown in Figure 3. Drugs are shown separately for each concentration given in μM . P-values are indicated for each comparison (t-test).

4) Even though factor 8 was significantly associated with time to next treatment (Figure 5A) and it explains some proportion of the gene expression variability, the factor shows no enrichment based on the GSEA (Figure 2E). Moreover, there is not much information / knowledge about what this factor is correlated to besides the one line in the text 'Factor 8 is associated with differences in gene expression between pre-treated and untreated patients'.

Can you elaborate more about Factor 8? For example provide a plot similar to Sup Figure 15?

We now discuss in more detail the set of genes that are most strongly associated with Factor 8, which is primarily active in the mRNA data. Of note, already in the initial submission, we reported a small set of gene sets that were enriched for genes linked to factor 8 (GSEA, **Fig.2e**), however these results were not discussed in the main text. We now elaborate on these findings, which link Factor 8 to Wnt signalling ($P=5 \times 10^{-05}$; FDR < 1%). See page 7, paragraph 3 and the new **Appendix Figure S19**.

5) The CLL variable is time to next treatment. It is very likely that individuals currently or proximally treated will have important changes in gene expression and that simply knowing if a person is being actively treated or not plus a few other clinical variables could be very helpful in predicting the timing of the next treatment. Particularly in the context of how CLL and its therapy may be timed in practice. This is highlighted in the observation that the response, next treatment timing, is best predicted by factors correlated to whether the patient was chemo-immunotherapy treated before collected. The relationship of the predictors to clinical variables such as these is not adequately explored and ultimately it is difficult to assess whether the molecular approach is better than an alternative approach that uses a combination of clinical phenotypes (like cell counts or further accounts for accelerated treatment for increased disease severity).

It is correct that clinical covariates, such as whether the patient was treated with chemo-immunotherapy before or the leukocyte counts as well as the cell doubling time, are known predictors of the time to next treatment (TTT). Clearly, these variables should be taken into consideration for clinical decision making. The primary aim of our analysis was to explore the molecular basis of clinical endpoints and the utility of MOFA for identifying relevant molecular factors, rather than improving upon the prediction of clinical endpoints based on classical markers.

Having said this, we agree that clinical variables are useful to provide context. We have now included this into the manuscript (paragraph 2, page 7), where we refer to an additional supplementary figure (**Appendix Figure S21**), comparing predictions based on MOFA factors with clinical covariates. This analysis supports the relevance of MOFA factors, for which we observe a similar predictive accuracy as for clinical covariates (cell doubling time and pretreatment) that are used to guide treatment decisions.

6) Is the data for the CLL example downloadable with MOFA?

The processed data as well as the trained MOFA model are contained in the vignette of the MOFAtools R package available in <https://github.com/bioFAM/MOFA>. The vignette illustrates the core functionalities provided by MOFA and enables reproducing the main findings reported in the paper.

The unprocessed raw data as presented in the original study can be obtained from <http://pace.embl.de/> (see Dietrich, Oles, Lu et al (JCI, 2018)).

Minor:

1) Figure S2c appears to show inflation of number of inferred factors when using binary data. Figure S3c is also inflated for Gaussian data. Can the authors comment on this?

It is true that in some cases, especially for non-Gaussian likelihoods, there is an inflation in the number of inferred factors. In general, identifying the correct number of factors is a challenging and still unsolved problem for factor models. In MOFA we approach this problem by combining the ARD prior with a variance threshold to inactivate factors during training (see **Methods**, *Model training and selection*). While this heuristic performs reasonably well and improves upon existing implementations of factor models (see the comparison with GFA in **Appendix Figure S4** and **Methods**, *Relationship to existing methods*), we agree that this strategy is not free of limitations. Finding more general solutions to determine model complexity in factor models remains an interesting problem for future research.

In MOFA much of the variation in the number of factors can be explained by the variational training converging to different local optima. Indeed, in the simulation study of non-gaussian likelihoods we observe that the inflated factor numbers correspond to poorer quality solutions. To address this, the MOFA implementation runs multiple restarts of our model, and the solution that maximises the evidence lower bound (ELBO) is selected. This training mode is used throughout the simulations as well as the applications, e.g. **Appendix Figure S6**.

In the revised manuscript we have altered **Appendix Figures S2** and **S3** by including the ELBO statistics. This demonstrates that solutions that maximizes the ELBO corresponds to the correct model complexity ($K=10$ factors), both for Gaussian and non-Gaussian likelihoods.

2) Page 3, second paragraph, add citations for the iCluster method after `Comparison of MOFA with the factor model that underlies the widely-used iCluster method (Shen et al 2009)` and other related approaches after `We contrast MOFA to related approaches in Methods and Supp. Table 1.`

We have added the references as suggested.

3) Page 3, second paragraph: ` We contrast MOFA to related approaches in Methods and Supp. Table 1`. The correct reference for this is "Supplemental Methods" and "Table 1 of Supplemental Methods" (not Supplemental Table 1 which refers to simulation analysis parameters).

We changed the reference, which is now **Appendix Table S3**.

4) All figure reference in "Supplemental Methods" are off by one number. In section 2.1, it's Figure S1 that contains results for recovery of the number of factors and not Figure S2. In section 2.2, it's Figure S2 and Figure S3 that contain results for non-Gaussian data and not Figure S3 and Figure S4 etc.

We apologize for this oversight and have fixed the references.

5) Figure S5b doesn't appear to show R² and colors on the same scale.

We modified the colors so that R² is shown on the same scale for all panels.

6) For Figure S7b can the authors comment on the trend for Factor 1?

We thank the reviewer for pointing this out. It was indeed an error, where Factor 1 corresponded to the intercept factor, which is constant across all different model instances (it is a vector of ones). The analysis was repeated and Figure S7b (now **Appendix Figures S7**) was updated accordingly.

Reviewer #2:

Argelaguet et al. present algorithms and code to implement group factor analysis for multi-omics data, which they term MOFA. They demonstrate the utility of MOFA on a public multi-omics data cohort, derived from leukaemia clinical samples.

Different variants of factor analysis, canonical correlation analysis and principal components analysis are widely used in systems biology studies. However, the authors have addressed some of the computational issues involved in applying group factor analysis to multi-omics data, including: code for fast inference, inference of sparse solutions, handling missing values, and likelihood models for integrating Boolean and count data.

The manuscript addresses issues encountered in applying group factor analysis to multi-omics data. The use of variational Bayes for fast inference, inference of sparse solutions and the handling of missing values have all been previously described. The use of Poisson and Negative Binomial likelihoods for factor analysis has been previously described. The analysis of CLL data is interesting and demonstrates the power of the approach, but these results are not followed up on in any way. The code is well written and well documented, and likely to be useful for bioinformatics practitioners with a strong statistical background. The manuscript is clearly written, the supplement provides an excellent primer on factor analysis for multi-omics, useful for readers with a strong mathematics background.

Major:

Although the code appears to be a careful implementation of group factor analysis for multi-omics data, and would likely be useful, it is not clear what the methodological advance of the study is. By their own admission, Table 1 Supp. 1.5, the advances of the study (as described in the introduction) have all been developed previously. Although not listed in that table, there are other implementations of Poisson and NB likelihoods too.

The application of MOFA to CLL data is interesting, and very carefully executed. But, there has been no follow-up studies performed. Thus, it is difficult to say that this manuscript represents a vertical advance in the field of CLL research.

Ultimately, the main contribution of the manuscript appears to be the code and the tutorial, which does seem very useful. I would ask the authors to consider either developing the CLL analysis through additional molecular assays, and refocus the paper on those results. Or, more clearly delineate how MOFA is a methodological advance over existing GFA approaches.

We thank the reviewer for these comments, which have helped to clarify the novelty of our paper. As suggested, we have shifted the focus of the manuscript to the methodological advances and practical advantages MOFA offers compared to existing methods. As part of these changes, we also present additional comparisons to existing group factor analysis models.

Briefly, the key advantages of MOFA include first, a more flexible and configurable noise model and improved computational efficiency as well as an efficient heuristics to estimate the numbers of factors. These differences are discussed in detail in *Model validation and comparison on simulated data* (page 4) and **Methods** 'Relationship to existing methods' section, where we also present empirical comparisons using simulated and real data. See also our response to Reviewer 1. While implementations of non-Gaussian factor analysis

have been proposed for single-view data, to our knowledge there is no group factor analysis model that can handle different data modalities. Among other multi-omics integration methods only iClusterPlus (Mo et al. 2012) is capable of handling non-Gaussian noise. We have included the iCluster methods into **Appendix Table S1**.

Additionally, in order to demonstrate that MOFA is widely applicable to different problems, we have included a second application to a multi-omics single-cell dataset where RNA expression and DNA methylation were simultaneously profiled in a set of 87 mouse ES cells. On this dataset, MOFA factors captured differentiation trajectories and yielded connections between transcriptomic and epigenetic changes.

Minor:

The title "Multi-Omics factor analysis disentangles heterogeneity in blood cancer", suggests that this is a manuscript about blood cancers. However, the only result for blood cancer I can find in the abstract is: "... [we] discovered previously underappreciated drivers of variation, such as response to oxidative stress.", which is somewhat of a modest result from which no conclusions are drawn. The manuscript itself reads like a methods paper, and in my opinion would make a strong software note. But, the title should reflect the ultimate direction the authors choose for the manuscript.

Thank you for this suggestion. In order to better reflect the methodological focus of the paper, we have changed the manuscript title to: "Multi-Omics factor analysis - a framework for unsupervised integration of multi-omic data sets"

Reviewer #3:

The manuscript presents an elegant Bayesian model "MOFA" based on group factor analysis to infer factors explaining variation across multiple omics datasets. There is a strong need for computational methods that allow integration of multiple data modalities such as expression, mutational and clinical data collected from the same system and this paper presents a principled methodology that provides interpretable factors for studying heterogeneity across samples from integration of datasets. The authors design a variational inference procedure that includes nice mean-field approximation for non-Gaussian data with two instructive examples for Bernoulli and Poisson data.

The factor space can be used for analysis and visualization, identification of outliers and imputing missing data. The authors show the performance on both simulated and CLL data. They identify the drivers of variation in CLL and molecular signatures based on integration of features and data modalities. They show that the inferred factors explain differences in expression, mutations, methylation or treatment outcome, etc.

This method is a very useful resource and tool for studying heterogeneity in a global scale through combination of matched data types. The model has the advantage to handle large amounts of clinical data, which is common.

I would very much like to see it published at MSB, but feel that some more analysis and interpretation is still needed, to make this into a better paper. See major comments below.

We thank the reviewer for the supportive comments on our manuscript.

Major comments

- Based on inferred factors on CLL patient data, the authors show MOFA's Ability of detecting outlier samples as well as imputing missing data in drug response. The simulation experiments showing the performance in imputing missing drug response data was great. However, I wasn't fully convinced with the outlier detection analysis. If the authors believe that some patients were mislabeled based on IGHV status (and were not borderline cases), they should examine what other markers or mechanisms are better surrogates of CLL biology and validate that experimentally.

The reviewer raises an important point, and we have followed their suggestion and tested the prediction experimentally. To recap, MOFA revealed three samples whose underlying molecular status appeared to be inconsistent with the reported IGHV status. This could be due either to a mislabeling/swapping of these samples; or it could point to interesting biological phenomena where the 'canonical' relationship between the molecular state of the cells and the IGHV status, as currently called from the DNA-sequence, does not hold.

We have generated additional data and extended our analysis to test the MOFA predictions for these three samples. We considered:

- (1) An independent drug response assay using a drug targeting the BCR pathway that was not contained in the dataset used for the training of MOFA (ONO-4509),
- (2) Whole exome sequencing (WES) data, at the locus of the IGHV genes.

Regarding (1), the drug sensitivity of sample P0108 indicated low reliance on BCR signalling, as is typical for M-CLL, despite its annotation as U-CLL, and in agreement with the MOFA prediction. For the other two samples, the drug responses were intermediate and

would be consistent both with their nominal IGHV status and the MOFA predictions. (**Figure EV3e**).

Regarding (2), the mutation load in the IGHV loci for all three cases was in agreement with the prediction by MOFA, i.e. for the two nominal U-CLL samples we found many mutations in the IGHV genes, while for the nominal M-CLL sample we found none (**Figure EV3f**).

In addition, we were able to commission an independent assessment of the IGHV status of sample P0108 from the attending clinician of the patient (Department of Haematology at University Hospital Mannheim; standard protocol for IGHV mutation analysis using PCR amplification and sequencing analysis of the IGHV region). They reported 93.3% nucleotide sequence homology in the V1-69 gene usage; given that the conventional cut-off is 98%, this report also supports the prediction of MOFA.

Taken together, these additional data and analyses confirm that the outlying samples identified by MOFA stand out functionally and that MOFA justifiably identified them as outliers: For P0108 both assays as well as the independent IGHV label support a reassignment to the IGHV group predicted by MOFA. For P0437 drug response assays in 1. are inconclusive but WES is strongly supportive of the IGHV group inferred by MOFA. For P0432 the results from both assays are consistent with either IGHV group but the lack of any mutations in WES is more common in U-CLL samples. While we have not identified the root cause underlying the fact that these three samples were outliers, we speculate that they may have been due to sample swapping during the assessment of the clinical IGHV label or (manual) data entry errors. We included these results on page 7, paragraph 1 of the manuscript and **Figure EV3**.

- Why are factors 3 and 4 not discussed in the paper? Factor 3 is also active in the drug response view, what does it represent?

We agree that Factors 3 and 4 received little attention in the previous version of the manuscript and we now provide more details on these factors (page 6, second paragraph and **Appendix Figure S14, S15**).

Factor 4 is mostly active in the mRNA data (9 % of variance explained) and the gene set enrichment analysis shows an enrichment of pathways related to immune response and T-cell receptor signalling (**Appendix Figure S14a**). Among the top weights of this factor are genes involved in the immune response, including surface markers such as CD300E, CD8A, CD3E (**Appendix Figure S14b,c**). This suggest that this factor captures differences in cell type composition between the samples: While most cells in the samples are B-cells, other cell types, such as T-cells and monocytes, are contained in varying amounts.

Factor 3, on the other hand, is mainly active in the drug response data (11% of variance explained). Notably, we observed that most drugs had positive weights on this factor (**Appendix Figure S15a**) and the factor is positively associated with the mean sensitivity of a sample across all drugs and concentrations (**Appendix Figure S15b**). Overall this suggests that the factor captures variation in the overall drug susceptibility (Geeleher et al, 2016).

- Supplementary Figure 11 shows Factor 1 when one view is masked, what are the biological signals that are missed with masking each view? Can you extend this to other factors?

In response to this comment, we have extended this analysis to all factors. Briefly, we trained MOFA excluding one of the views at a time. We then compared the factors inferred on the reduced dataset to those from the full dataset (using the maximal correlation coefficient across factors). This approach allowed for quantifying to what extent the reduced MOFA model captures original factors inferred on the full dataset (**Appendix Figure S8c**).

Overall, we observed that a factor active in the masked view can still be recovered, provided that it is active in at least one other view. As already previously demonstrated Factor 1 is always recovered, which is consistent with its broad sharing across all views. A similar trend can be observed for Factor 2. Other factors are reliant on the mRNA and methylation data and hence cannot be identified without the corresponding assay. Surprisingly, we also find some factors that had strong activities in the drug response data (such as Factor 3 and Factor 5) to be heavily relying on the inclusion of other assays, such as mRNA or methylation. This could be due to a higher noise level in the drug response data that can be mitigated by combining with other assay or due to a low number of features, for which this factor explains variation in the drug response assay compared to other assays.

- Also, how would the results look like if only one view was used (and the rest were masked)? Which views were most informative of clinical outcome or CLL markers?

The reviewer touches upon an important point that has not received sufficient attention in the previous version of the manuscript. Running the MOFA model with a single view is equivalent to a Bayesian Sparse Factor analysis, which is intuitively related to a (sparse) Principal Component Analysis but with an explicit probabilistic formulation.

In our application we observed that many of the factors can in principle also be inferred based on a single view, with the mRNA data being most informative overall (**Appendix Figure S8a**). A single view model, however, does not yield connections between the different views. A second advantage of the multi-view approach is that the full dataset can be leveraged. For example, on the CLL data MOFA is applied to 200 samples, whereas there are only 136 samples with mRNA data.

In terms of information for clinical outcome, we again found that the mRNA data to be the most informative single view (**Appendix Figure S8b**). To ensure a fair comparison in this analysis, we only considered the subset of samples for which all data modalities were profiled.

- Overall, while the computational methodology is strong, the paper does not show MOFA's ability in inferring deep biological insight. Therefore, I suggest either performing deeper analysis and biological experiments or applying this method to a second system such as ENCODE datasets to show broader application and insight.

First, we would like to note that the revised manuscript now places stronger emphasis on the methodological advances of MOFA and less on a specific findings in CLL biology. This change addresses comments raised by reviewers 1 & 2, as well as editorial requests.

Having said this, we agree that a compelling application of the method is important. In response to referee comments, we have extended the analysis on the CLL data, for which we provide further details (such as discussed in response to the comments 1 and 2, as well as Reviewer 1 comments 4 and 5). Second, in order to demonstrate the broad applicability of the model, we now present a second application of MOFA to a single cell multi-omics dataset where RNA expression and DNA methylation were simultaneously profiled in a set of 87 mouse ES cells. Here, MOFA yielded insights into coordinated changes between the transcriptome and the epigenome along a differentiation trajectory (page 7-8, new **Figure 5**).

Minor points

- Supplementary methods: Section 1.1 refers to Figure S1 for plate model, should be Figure S17.

Thanks for pointing this out, we changed the reference (now **Appendix Figure S24**).

- Figure 2E: Were no gene sets enriched for factor 1,2?

With the Reactome gene set, which we used in the current manuscript, no gene sets were enriched for Factors 1 and 2. Depending on the study of interest other gene sets can be used in the gene set enrichment analysis function implemented in MOFA. In our CLL application, using leukemia specific gene sets or positional gene sets for example would connect these two factors to B-cells activity and chromosome 12, respectively. In general, we leave the gene sets to be tested as a user choice in MOFA.

- Figure 2F not cited in main text or explained. Some subfigures are not cited in order.

We now refer to this Figure at the end of page 4, where we mention the relationship of Factor 1 and 2 with IGHV status and Trisomy 12. Panels were re-ordered to match the order in the manuscript.

- Figure 4b: Was the gene set enrichment done between the two factor clusters (marked in Figure 4a)? The samples don't show discrete clusters for factor 5 unlike factor 1. What is the significance of clustering factor 5 into two clusters?

The reviewer is right that the clustering of Factor 5 in two groups is arbitrary. In general, our analysis is always based on the continuous factors and their weights, e.g. for the gene set enrichment analysis we used the weights of the continuous factors (**Methods**). Only for visualization purposes did we sometimes opt for a discrete clustering. However, one of the advantages of MOFA is that no discrete clustering is required. Working with the continuous factor is in many cases a better representation of the underlying molecular phenotype than grouping samples in discrete subpopulations. Therefore, we updated Figure 4 (now **Figure EV2**) with a continuous representation of the samples along the factor.

- Page 5 second paragraph from bottom: reference should be to Fig 2f not Fig 2d.

We corrected this reference, supposed to refer to Figure 2e (gene set enrichment analysis).

- Fig S9: Would be great to show the sample number breakdown by views, to see if factor clusters are associated with (or biased to) views.

We added the sample number breakdown by views to Figure S9 (now **Appendix Figure S11**) and did not find any association of the sample numbers per view to the clusters.

Thank you for sending us your revised manuscript. We have now heard back from the two reviewers who were asked to evaluate your manuscript. As you will see below, the reviewers mention that their concerns have been addressed and think that the study is now suitable for publication. Reviewer #1 mentions a couple of minor issues with the text, which we would ask you to fix.

REFEREE REPORTS.

Reviewer #1:

The authors have well addressed the comments.

Minor issues:

Two typos, both in section "Relationship to existing methods" p. 13 : (1) "iCluster: In contrast to MOFA, iCluster uses in *a* each view" -> "iCluster: In contrast to MOFA, iCluster uses in each view" and (2) "Group factor analysis: While the underlying model of MOFA is closely *connect* to" -> "Group factor analysis: While the underlying model of MOFA is closely *connected* to"

Reviewer #3:

The manuscript has significantly improved and all the main comments from all reviewers have been addressed. The addition of single-cell data (despite the small scale) makes a strong distinction between this method and previous methods such as iCluster in that MOFA provides model flexibility to accommodate both continuous and discrete states. The main novelty of the paper as a first flexible and configurable group Factor Analysis method for multiple data modalities is now clear. Adequate comparisons to previous factor analysis methods have been added in the revised manuscript.

It would be interesting if the authors applied this method to single cell data showing hierarchical differentiation with multiple branching trajectories such as hematopoietic differentiation, though a large-scale application might be beyond the scope of this paper.

The addition of validation data (drug response and WES) has strengthened the method, but still does not provide new biological insight. In this regard, repackaging the paper as a methodology paper rather than focusing on novel CLL insight is befitting.

The flexibility of the noise model would expand the possible applications of this method. Furthermore, the supplementary provides a valuable resource (and a tutorial) for statistical computational biologists.

Editorial comments:

- Please include five keywords and a running title.
multi-omics, data integration, factor analysis, dimensionality reduction, single-cell omics.
- We noticed that you have not filled boxes 20 and 21 of the author checklist, could you please include an answer for these two boxes?
Done.
- In the main text: please include callouts to Figure panels 1A and 1B.
Done.
- In the Appendix PDF, we would ask you to rename the Figures from Supplementary Figure SX to Appendix Figure SX.
Done.

Reviewer comments:**Reviewer #1:**

The authors have well addressed the comments.

Minor issues:

Two typos, both in section "Relationship to existing methods" p. 13 : (1) "iCluster: In contrast to MOFA, iCluster uses in *a* each view" -> "iCluster: In contrast to MOFA, iCluster uses in each view" and (2) "Group factor analysis: While the underlying model of MOFA is closely *connect* to" -> "Group factor analysis: While the underlying model of MOFA is closely *connected* to"

Thank you for the positive comment. We have addressed all these comments and typos.

Reviewer #3:

The manuscript has significantly improved and all the main comments from all reviewers have been addressed. The addition of single-cell data (despite the small scale) makes a strong distinction between this method and previous methods such as iCluster in that MOFA provides model flexibility to accommodate both continuous and discrete states. The main novelty of the paper as a first flexible and configurable group Factor Analysis method for multiple data modalities is now clear. Adequate comparisons to previous factor analysis methods have been added in the revised manuscript.

It would be interesting if the authors applied this method to single cell data showing hierarchical differentiation with multiple branching trajectories such as hematopoietic differentiation, though a large-scale application might be beyond the scope of this paper.

The addition of validation data (drug response and WES) has strengthened the method, but still does not provide new biological insight. In this regard, repackaging the paper as a methodology paper rather than focusing on novel CLL insight is befitting.

The flexibility of the noise model would expand the possible applications of this method. Furthermore, the supplementary provides a valuable resource (and a tutorial) for statistical computational biologists.

Thank you for the positive comments. We agree that the results we present can be extended in a number of ways. In particular, applications to larger single-cell data sets will be fruitful directions in the future.

Corresponding Author Name: Oliver Stegle

Manuscript Number: MSB-17-8124R